# Drug Repositioning in Doxorubicin-Induced Cardiotoxicity Protection

**DOI:** 10.3390/ijms262010130

**Published:** 2025-10-17

**Authors:** Marija Kosić, Vladislav Pajović, Mirjana Jovanović, Nina Japundžić-Žigon

**Affiliations:** 1Institute of Pharmacology, Clinical Pharmacology and Toxicology, Faculty of Medicine, University of Belgrade, 11000 Belgrade, Serbia; marija.kosic@med.bg.ac.rs (M.K.); vladislav.pajovic@med.bg.ac.rs (V.P.); 2Institute of Pathophysiology, Faculty of Medicine, University of Belgrade, 11000 Belgrade, Serbia; jovanovic.mirjana@yahoo.com

**Keywords:** doxorubicin, cardiotoxicity, drug repurposing, cardioprotection, paroxetine

## Abstract

Doxorubicin (DOX) is an effective drug for the treatment of solid tumors and hematological malignancies in both children and adults. The most serious side effect is doxorubicin-induced cardiotoxicity (DIC), which can lead to cardiomyopathy and irreversible and highly fatal cardiac decompensation. The precise mechanisms underlying DIC are not fully understood, and currently, no fully effective preventive or therapeutic strategies exist. Drug repositioning has emerged as a promising approach to mitigate DIC, leveraging existing safety profiles while potentially reducing the time and cost of clinical translation. In this review, we summarize current evidence on drug repurposing for DIC, with a particular focus on the antidepressant paroxetine, which shows potential cardioprotective effects beyond its established role as a selective serotonin reuptake inhibitor (SSRI).

## 1. Introduction

Anthracyclines remain a cornerstone of chemotherapy for both solid tumors and hematologic malignancies, even though targeted therapies and immunotherapies have significantly improved survival in many cancer patients [1]. Doxorubicin (DOX), the most commonly used anthracycline, was first isolated from the *Streptomyces peucetius* bacterium [2]. The anticancer activity of DOX is mainly based on its ability to intercalate DNA, inhibit topoisomerase 2 (TOP2), and induce apoptosis [3]. However, its therapeutic potential is limited by a significant risk of life-threatening cardiotoxicity [3].

The problem of DOX-induced cardiotoxicity (DIC) has become more evident in recent years, largely due to an aging population, more patients with chronic comorbidities, and improved long-term cancer survival. In anthracycline-treated patients, a nine-year follow-up reported 17.9% subclinical and 6.3% overt cardiotoxicity [4]. DIC is either acute or chronic. Acute toxicity develops during treatment or shortly after drug administration, most often presenting with transient electrocardiographic changes and is responsive to treatment [5]. Chronic DIC, on the other hand, typically appears within the first month after therapy but can also manifest years, or even decades, later—especially in childhood cancer survivors [6]. This form is usually irreversible (type 1 drug-induced cardiotoxicity), dose-dependent, and often progresses to dilated cardiomyopathy with subsequent heart failure (HF) [7].

Although the mechanisms underlying DIC have been studied extensively, the precise molecular pathways remain unclear [8]. Early diagnosis is often challenging, as cardiac injury can persist clinically silent for a long time, and conventional diagnostic tools, such as non-invasive echocardiography, lack sensitivity [9]. Preventive strategies are limited, and once cardiotoxicity becomes established, therapeutic options are scarce and ineffective [5]. Patients who develop congestive HF have a poor prognosis, with a 3-year survival rate of approximately 50% [10].

Given the significant challenges in timely diagnosis and the lack of effective preventive or therapeutic options for DIC, drug repositioning (repurposing, reprofiling, or re-tasking) has emerged as a promising new strategy. This relatively new concept, first defined by Ashburn and Thor in 2004, refers to finding new uses for existing drugs [11]. It is based on the observation that many diseases share common biological targets and that most drugs exert multiple, pleiotropic effects through interactions with different molecular pathways [12].

The repositioning of drugs greatly simplifies the regulatory procedures for market approval and reduces both the time and cost of research, since the safety profile and toxicity of the drug are already well established [13]. However, any modification in formulation, dosage, or route of administration will require a re-examination under new conditions [12]. Moreover, the increased availability of public databases and computational tools has significantly facilitated the identification of new drug candidates for repositioning [14].

The literature search primarily covered studies published in the past 15 years. Earlier works, such as the first original contributions in the field, were included to provide historical context and highlight findings that remain relevant for understanding DIC mechanisms. A PubMed search on DIC revelead 297 review articles for a period of five years (https://pubmed.ncbi.nlm.nih.gov/?term=doxorubicin+cardiotoxicity, accessed on 28 September 2025). In this paper, we provide, to the best of our knowledge, the first comprehensive review that integrates the mechanisms of DIC with the potential role of drug repurposing, highlighting paroxetine as a promising candidate that may exert protective effects beyond its known role as an antidepressant medication of the selective serotonin reuptake inhibitor (SSRI) class [15]. Considering that depression affects more than 10% of cancer patients [16] and occurs at rates up to three times higher than in the general population [17], paroxetine may provide a dual therapeutic benefit, addressing both cardiotoxicity and mental health.

## 2. Molecular Mechanisms of Doxorubicin-Induced Cardiotoxicity: Potential Targets for Prevention

Several molecular mechanisms have been proposed for the development of DIC, with oxidative stress in cardiac tissue considered the classical pathway [18]. The cardiotoxic effect is based on similar mechanisms by which DOX exerts its desired antitumor effect: genesis of free radicals, formation of complexes with iron, cell membrane damage, disruption of intracellular ion transport, perturbations in the cellular energy status, subsequent activation of signaling pathways that lead to cell death, and disturbances of myocardial adrenergic signaling [19,20].

Cardiac cells are thought to be particularly vulnerable to oxidative damage and anthracycline toxicity due to the expression of the isoenzyme TOP2β, a primary binding target of DOX [21]. In addition, mitochondria, the main organelles of redox metabolism and cellular energy, constitute nearly 50% of cardiomyocyte mass [22], while the activity of antioxidant enzymes in cardiac tissue is relatively low [23]. Positively charged DOX shows a high affinity for negatively charged cardiolipin, a phospholipid of the inner mitochondrial membrane, making mitochondria the most extensively damaged subcellular organelles in DIC [20].

### 2.1. Oxidative Stress

Excessive generation of reactive oxygen species (ROS) occurs through multiple pathways, including enzymatic reactions involving NADPH oxidase (NOX) and nitric oxide synthase (NOS), mitochondrial ROS production, and the formation of iron–DOX complexes. These processes synergistically contribute to redox imbalance, disruption of cellular homeostasis, and irreversible cardiac injury [8].

NOX2 plays a central role by reducing the quinone moiety of DOX to a semiquinone radical, thereby initiating a cascade of ROS formation [24]. Genetic polymorphisms in NOX subunits have been associated with differences in susceptibility to acute and chronic forms of cardiotoxicity, suggesting potential value as predictive biomarkers [25].

Another important pathway involves nitric oxide synthases (NOS). DOX binds to endothelial NOS (eNOS), leading to the formation of semiquinone radicals and an imbalance between superoxide and nitric oxide in favor of superoxide, which exacerbates cardiac tissue injury [20]. Simultaneously, induction of inducible NOS (iNOS) further amplifies nitrosative stress [26].

Mitochondria are also a major source of DOX-induced ROS. Binding of the drug to mitochondrial cardiolipin disrupts respiratory chain function and promotes superoxide generation, while concomitant damage to mitochondrial DNA and proteins aggravates oxidative injury [27,28]. These changes impair fatty acid oxidation, shift energy metabolism toward glucose utilization, and compromise cardiomyocyte contractility [29].

The formation of DOX–iron complexes catalyzes the conversion of less reactive species into highly toxic hydroxyl radicals. At the same time, DOX disrupts intracellular iron homeostasis by altering the activity of iron regulatory protein (IRP), thereby affecting ferritin and transferrin receptor function and further promoting iron-driven ROS generation [22,30].

### 2.2. Induction of Apoptosis and Ferroptosis

DOX activates both the intrinsic and extrinsic apoptotic pathways [8]. The intrinsic pathway is activated by internal cellular stress, such as DNA damage, while the extrinsic pathway is initiated by external signals from other cells, typically via death receptors. DOX upregulates death receptor expression and enhances p53-mediated proapoptotic signaling [31].

Ferroptosis is a novel form of regulated cell death driven by increased cardiac iron levels, loss of antioxidant defenses, and lipid peroxidation [32]. Recent research has suggested ferroptosis as both a key contributor to DIC pathophysiology and a promising therapeutic target. Iron chelators, radical-trapping antioxidants, glutathione precursors, and medications targeting heme metabolism have been shown to alleviate myocardial injury by targeting ferroptotic pathways [33].

### 2.3. Disruption of Calcium Homeostasis

DOX perturbs intracellular calcium handling, leading to calcium overload, impaired contractility, and activation of apoptotic pathways [8]. Its metabolite, doxorubicinol, interferes with the Na^+^/Ca^2+^ exchanger, while ROS-mediated alterations of calcium-regulating proteins further exacerbate calcium imbalance. Collectively, these changes promote mitochondrial dysfunction and cardiomyocyte death [34,35].

### 2.4. AMPK Signaling Pathway

DOX has been shown to inhibit the adenosine monophosphate-activated protein kinase (AMPK) signaling pathway, reducing the enzymatic activity of acetyl-CoA carboxylase (ACC). Inhibition of this pathway activates protein kinase B and mitogen-activated protein kinase (MAPK), contributing to DNA damage, metabolic stress, and cardiac hypertrophy [8,36]. These findings highlight AMPK as a possible target for cardioprotection in DIC.

### 2.5. Role of Endothelin-1

DOX increases the level of endothelin-1 (ET-1) in cardiomyocytes and leads to hypertrophic cardiomyopathy via the epidermal growth factor (EGF) receptor signaling pathway [8]. Antagonists of endothelin A and B receptors show cardioprotective effects by reducing the level of tumor necrosis factor alpha (TNFα) and mitigating lipid peroxidation [37].

### 2.6. Extracellular Matrix Remodeling and Inflammation

In tumor cells, DOX inhibits the synthesis of collagenases and matrix metalloproteinases (MMPs), enzymes that degrade the extracellular matrix and facilitate tumor expansion. In contrast, in cardiac tissue, DOX exerts the opposite effect on MMP-2 and MMP-9, leading to increased collagen type I and type III deposition and myocardial fibrosis [38]. Fibrosis is further driven by DOX-induced activation of cardiac fibroblasts, increased expression of profibrotic cytokines such as transforming growth factor-beta (TGF-β), and excessive deposition of extracellular matrix proteins, ultimately leading to impaired left ventricular (LV) contractility [39,40]. This remodeling is permanent, resulting in myocardial dysfunction and HF [41].

Inflammation represents an important mechanism underlying DIC. DOX activates intrinsic immune pathways, particularly through Toll-like receptors (TLRs), which stimulate the release of pro-inflammatory cytokines, including TNF-α, interleukin-6 (IL-6), and interleukin-1 beta (IL-1β). This amplifies the inflammatory response and contributes to cardiomyocyte damage [42]. A central mechanism involves activation of the NLRP3 inflammasome, which triggers caspase-1–dependent pyroptosis, a pro-inflammatory form of cell death [43]. Pyroptosis further exacerbates cardiac injury by releasing cytokines and damage-associated molecular patterns (DAMPs), thereby promoting inflammation and myocardial fibrosis [44].

Taken together, targeting these inflammatory and profibrotic cascades may represent a promising therapeutic strategy to mitigate DIC [41,42]. For example, modulation of TLR4 signaling has shown cardioprotective effects by lowering pro-inflammatory cytokine levels and preserving cardiac function [44], while DOX-induced cardiac fibrosis can be attenuated through neurokinin-1 receptor blockade [45].

### 2.7. Disturbances of Myocardial Adrenergic Signaling

A hallmark of DIC is the functional remodeling of beta-adrenergic receptors (β-ARs), key mediators of cardiac inotropism [46]. Early studies demonstrated reduced β-AR affinity for noradrenaline and impaired adrenergic signaling, contributing to cardiac decompensation [47]. Conflicting results have been reported for β-AR density, subtype distribution, and functional coupling, depending on the species, route of DOX administration, dosing, and, in particular, the stage of DIC [48,49,50,51,52]. Clarifying these alterations is clinically relevant, given their implications for the use of β-AR antagonists and novel modulators of adrenergic signaling in DIC.

An important mechanism of β-ARs density regulation involves the phosphorylation by specific receptor kinases, the G protein-coupled receptor kinases (GRKs), and binding of beta-arrestins targeting β-ARs for clathrin-mediated internalization and intracellular degradation [53]. GRK2 is up-regulated in experimental models of HF and is involved in the development of cardiac hypertrophy [54]. Inhibition of GRK2 has been proposed as a new therapeutic target in HF with a potentially significant impact on the progression and outcome of this disease [53].

### 2.8. Epigenetic Changes in Doxorubicin-Induced Cardiotoxicity

Epigenetic mechanisms, including DNA methylation, histone modifications, and non-coding RNAs, have been increasingly implicated in the development of DIC [8,55,56]. These modifications alter gene expression without altering the DNA sequence and can be influenced by age, environment, and disease [8].

Methylation of cytosine-guanine dinucleotides is a major epigenetic mechanism responsible for the silencing of 60–90% of human genes [57]. Doxorubicin has been shown to downregulate DNA methyltransferase 1, resulting in global hypomethylation and dysregulation of mitochondrial genes [55].

DOX alters the chromatin landscape and gene expression related to cardiac function by inhibiting histone deacetylases. This inhibition activates Rac1, NOX, and p53 signaling pathways, ultimately promoting cardiomyocyte apoptosis and hypertrophy [8]. In addition, histone-modifying enzymes contribute to chemoresistance by upregulating anti-apoptotic genes, thereby reducing drug-induced cell death [58].

Several microRNAs are dysregulated in DIC, with miR-23a, miR-34a, miR-140, miR-146a, miR-532, and miR-15 upregulated, and miR-29b and miR-30 downregulated [59]. These changes affect cardiomyocyte survival, electrical conduction, and contractility; for instance, miR-15 upregulation promotes apoptosis in H9c2 cells [60].

### 2.9. Cardiotoxicity Beyond Cardiomyocytes: Fibroblasts and Other Cellular Targets of Doxorubicin

Although cardiomyocytes have long been the main focus of anthracycline cardiotoxicity research, they represent less than one-third of all cardiac cells. Growing evidence shows that non-myocyte populations are equally important [7]. Among them, cardiac fibroblasts stand out as the principal cells that mediate cardiotoxic effects of DOX. DOX induces their transformation into myofibroblasts, boosting extracellular matrix production and fibrosis [61]. It also activates ATM-dependent Fas ligand release in fibroblasts, which in turn triggers apoptosis in neighboring cardiomyocytes [62].

DOX further compromises cardiac progenitor cells. By inducing ROS-dependent DNA damage, it promotes cellular senescence and apoptosis, ultimately impairing myocardial regeneration [63]. Endothelial cells are another direct target: DOX provokes oxidative stress, calcium overload, and eNOS dysregulation, all of which contribute to vascular dysfunction [64].

The effects extend beyond the heart. DOX damages bone marrow–derived mesenchymal stem cells, depleting systemic repair capacity. Mesenchymal stem cells isolated from animals treated with DOX showed a markedly reduced proliferation rate and a limited ability to respond to cardiomyogenic stimuli [65].

Taken together, these findings highlight that fibroblasts and other non-cardiomyocyte cell types critically contribute to the pathophysiology of DIC and may represent novel targets for protective strategies.

## 3. Current Strategies for Doxorubicin-Induced Cardiotoxicity Protection

Primary prevention of DIC is the best strategy to prevent “today’s cancer patients from becoming tomorrow’s cardiac patients [19].” Before starting therapy, it is essential to take a detailed medical history of the patient, with a focus on risk factors, existing cardiovascular diseases, and previous use of antitumor medications, as well as an assessment of cardiac function [66]. The overall benefit of DOX treatment should be evaluated against the potential risk of cardiotoxicity [19].

Secondary prevention refers to patient monitoring during and after treatment, enabling early recognition of cardiotoxicity and prompt initiation of therapy [9]. Although various guidelines have been published for monitoring cardiotoxicity in cancer patients, the effectiveness of specific regimens has not been established, and recommendations from different groups are inconsistent [1,9,67].

### 3.1. Dose Reduction

Since the incidence of chronic DIC is primarily dose-dependent, limiting the cumulative dose of DOX to 400–450 mg/m^2^ represents the first line of prevention [68]. However, as late cardiotoxic effects of DOX also develop in patients receiving doses below this level, there is probably no safe dose of antracycline [7]. Moreover, underexposure of malignant cells to cytotoxic drugs may affect their efficacy by triggering intracellular mechanisms for drug resistance [69].

### 3.2. Pharmacokinetics/Pharmacodynamics Approach

DOX displays a triphasic plasma clearance with rapid distribution into highly perfused tissues, followed by slow elimination through hepatic metabolism and biliary excretion [70]. Its tendency to accumulate in cardiac tissue, due to high affinity for cardiolipin in the inner mitochondrial membrane, contributes to persistent oxidative stress and delayed cardiotoxicity [71]. Moreover, the interpatient variability in DOX metabolism, influenced by genetic polymorphisms in drug transporters and metabolizing enzymes, may further modulate individual risk [72,73,74]. These pharmacokinetic aspects highlight the narrow therapeutic window of doxorubicin and emphasize the importance of developing cardioprotective strategies that do not interfere with its anticancer activity.

While the antitumor efficacy of anthracyclines depends on cumulative exposure, cardiotoxic effects are more likely linked to peak plasma concentrations [7]. Accordingly, preventive strategies include replacing bolus administration with slow infusion and using liposomal instead of conventional formulations [75].

DOX analogs, including epirubicin, idarubicin, amrubicin, and pixantrone, offer a chemotherapeutic alternative by modifying the DOX structure to reduce cardiotoxicity while retaining antitumor effects, though often with decreased potency compared to the original drug [76,77].

### 3.3. Nanosized Particle Approach

In the last few decades, nanodrug carrier systems have been developed for the targeted delivery of drugs, including DOX (Figure 1) [78,79,80]. A number of nanodrug carrier systems have been approved by the Food and Drug Administration (FDA). This new approach seems to be the most promising to increase antitumor activity while reducing cardiotoxicity. Nano carriers are biocompatible macromolecules with low immunogenicity that evade renal filtration and achieve long blood circulation time. They increase drug delivery through the enhanced permeability and retention (EPR) effect, thereby improving tumor targeting and reducing non-specific organ toxicity [81]. Furthermore, surface modifications with targeting moieties such as antibodies, carbohydrates, and other ligands can promote selective and efficient cellular uptake of nanoparticles [82]. In our experiments, conjugation of doxorubicin to N-(2-hydroxypropyl) methylacrylamide copolymer (HPMA) has significantly reduced the DOX-associated cardiotoxicity [83,84]. Indeed, accumulated evidence over the years indicates that pegylated-liposomal doxorubicin is associated with a low risk of DOX-induced HF [85]. However, its clinical use has revealed new and serious adverse effects, such as unpredictable and delayed dermal toxicity, most notably palmar–plantar erythrodysesthesia [86].

### 3.4. Dexrazoxane

Dexrazoxane is the only cardioprotective agent approved by the FDA in pediatric and adult patients exposed to DOX cumulative dose known to induce DIC [7]. It decreases ROS production by chelating iron [87]. Recently, it has been demonstrated that dexrazoxane can compete for the adenosine triphosphate (ATP)-binding site of TOP2β, thus preventing the formation of DOX-TOP2β complex and the inhibition of DNA replication in cardiac cells [21]. However, the clinical use of dexrazoxane has been limited following a few reports of its possible interference with the antitumor activity of anthracyclines, and the potential risk of secondary tumors, myelosuppression, and infection [7,19].

### 3.5. Antioxidants

Early studies on synthetic drugs or natural compounds with antioxidant properties, such as vitamins A and E, coenzyme Q10, probucol, curcumin, and allicin, have shown limited efficacy in improving DIC, suggesting that oxidative stress is not the key mechanism that mediates DIC [7,8]. Although some of these molecules do not deprive the antineoplastic efficacy, they did not provide substantial improvements to heart function [88].

### 3.6. Cardiovascular Drugs

Whether cardiovascular drugs, such as β-AR antagonists, angiotensin-converting enzyme inhibitors (ACEIs), angiotensin receptor blockers (ARBs), and mineralocorticoid receptor antagonists, are effective for the primary prevention of DIC remains questionable [89]. Vasodilatory β-AR antagonists with antioxidant activity, carvedilol and nebivolol, have shown protective effects in preclinical studies [1,7]. Most clinical studies on the primary prevention of LV dysfunction with these drugs had small samples and short follow-up after discontinuation of therapy. The limited beneficial and potential long-term adverse effects do not support their routine in DIC cardioprotection [1].

In contrast, there is evidence of clinically beneficial effects of ACEIs and β-AR antagonists when used for secondary prevention in patients with subclinical DIC. Early introduction of cardioprotective therapy is associated with higher chances of myocardial function recovery and fewer cardiac events, including arrhythmias, HF, and death [90].

### 3.7. Sensitive and Specific Methods for Early Diagnosis

The early detection of subclinical forms of DIC and the timely discontinuation of DOX can prevent irreversible cardiac damage [91]. In clinical practice, echocardiography is the method of choice for detecting myocardial dysfunction before, during, and after cancer therapy [92]. Elevated plasma levels of B-type natriuretic peptide (BNP) and cardiac troponins signal myocardial injury, but they lack both specificity and sensitivity for DIC [5].

The advent of advanced cardiac imaging techniques, including cardiac magnetic resonance (cMRI) and strain analysis by echocardiography, has revealed that the incidence of DIC is considerably higher than earlier estimates based solely on echocardiography [93]. However, all indicators of LV function provided by imaging modalities are poor predictors of chronic DIC after therapy [94].

Our experimental findings imply that comprehensive heart rate variability (HRV) and blood pressure variability (BPV) analysis might be useful for early diagnosis of DIC [48]. They suggest that subjects in the early stage of DIC, before the development of overt HF, show increased HRV due to transient up-regulation of cardiac β1-AR, and this can be prevented by the use of selective β1-AR antagonists [48,95].

### 3.8. Understanding of the Phenotypic Heterogeneity of Doxorubicin-Induced Cardiotoxicity

Current guidelines define cardiotoxicity primarily as LV dysfunction. However, the threshold for a clinically significant decline in LV ejection fraction varies with the measurement method and the specific guideline [96]. Importantly, these definitions do not capture the broader spectrum of DIC, which can present as diastolic dysfunction, atherosclerotic disease, or arrhythmia [1].

Experimental data show that DOX induces two DIC phenotypes in rats—one with reduced LV ejection fraction and one with preserved LV ejection fraction—occurring at similar incidence and with comparable detrimental outcomes [49]. While pharmacotherapy improves survival only in patients with systolic dysfunction, there are no effective treatments for HF with preserved ejection fraction, and mortality remains high [1,5]. Comprehensive studies of the cardiac transcriptome in both DIC phenotypes may reveal new targets for effective cardioprotection [49].

### 3.9. Pharmacogenomics

Pharmacogenomics, which examines genetic determinants of drug response, is central to personalized medicine and may help balance DOX efficacy and toxicity. Several polymorphisms in anthracycline metabolism and cardiomyopathy pathways have been proposed as predictors of DIC, but current evidence is limited and requires further validation [97].

## 4. Drug Repositioning in Doxorubicin-Induced Cardiotoxicity

Multiple agents with well-established safety profiles have been investigated for potential cardioprotective effects in DIC. In the following text, we outline the most studied candidates and summarize the current evidence for their repositioning (Figure 2, Table 1).

### 4.1. Statins

Statins, competitive inhibitors of 3-hydroxy-3-methylglutaryl-coenzyme A (HMG-CoA) reductase, were originally developed to reduce cholesterol synthesis in the treatment of hyperlipidemia [98]. Beyond lipid-lowering, they exert a wide range of ‘pleiotropic effects’, including anti-inflammatory, antifibrotic, immunomodulatory, antioxidant, vascular endothelial protective, and plaque-stabilizing effects [99].

Statins lessen oxidative damage in cardiac tissue by downregulating NOX-1 in vascular smooth muscle cells, reducing ROS production, and upregulating ROS-scavenging enzymes [98,100]. Their anti-inflammatory effect primarily stems from inhibiting the nuclear factor kappa B (NF-kB) signaling, thereby suppressing cytokine and adhesion molecule expression [101]. They also attenuate fibrosis by decreasing collagen expression and fibroblast proliferation, thus mitigating myocardial hypertrophy [102]. It has been reported that they activate AMPK, which enhances eNOS activity and contributes to cardiovascular protection [103].

In recent years, numerous preclinical and clinical studies have explored the role of statins in alleviating anthracycline-associated cardiotoxicity [98]. Feleszko et al. reported that lovastatin enhanced the antitumor efficacy of DOX while significantly reducing troponin T release from the cardiomyocytes of DOX-treated mice [104]. Consistent findings from other studies demonstrated that lovastatin and atorvastatin attenuated DNA damage through decreasing caspase-3-mediated apoptosis and cardiac inflammation, without compromising DOX therapeutic effects [105,106].

In a retrospective clinical study of women with breast cancer receiving DOX, those treated with statins exhibited a lower incidence of HF compared with the matched controls [107]. On this basis, the 2022 European Society of Cardiology (ESC) cardio-oncology guidelines recommend statins for primary prevention in patients at high risk of DIC [108]. Many studies have shown the benefits of statins for HF with preserved ejection fraction. On the other hand, two large clinical trials have produced controversial results regarding their effects on HF with reduced ejection fraction [109]. Future cardio-oncology research should therefore clarify whether statins exert differential protective effects across cardiotoxicity subtypes, enabling more targeted preventive strategies.

### 4.2. Metformin

Metformin is an oral biguanide that has been widely used as the first-line treatment for type 2 diabetes due to its safety, efficacy, and tolerability [110]. In patients with diabetes, it suppresses hepatic gluconeogenesis and enhances insulin-stimulated glucose uptake in skeletal muscle cells and adipocytes. These beneficial effects are achieved primarily through the activation of AMPK, a key regulator of mitochondrial homeostasis and cellular energy metabolism [111].

Metformin has been proposed to counteract DIC through several mechanisms, including attenuation of oxidative stress, preservation of mitochondrial function, normalization of autophagy markers, and activation of AMPK [112]. Although AMPK is generally considered cardioprotective, the effects of DOX on its activity appear variable, influenced by treatment dose, duration, and experimental model [36]. Moreover, activation of specific AMPK isoforms or holoenzymes can even be detrimental to the heart, underscoring the complexity of its isoform-dependent functions [112].

In various preclinical models of cardiac disease, metformin has been shown to lower the risk of HF and reduce cardiovascular mortality, independent of its glucose-lowering effects [113,114]. Its favorable safety profile, pleiotropic mechanisms of action, and low cost make metformin an attractive candidate for repurposing in DIC cardioprotection [115]. Nevertheless, most studies supporting metformin in mitigating DIC are preclinical, with limited evidence from translational clinical trials. A phase II clinical trial entitled “Use of Metformin to Reduce Cardiac Toxicity in Breast Cancer” (NCT02472353) was unfortunately terminated prematurely due to insufficient participant enrollment [110].

### 4.3. Sodium-Glucose Cotransporter-2 (SGLT2) Inhibitors

Sodium-glucose co-transporter 2 (SGLT2) inhibitors, including canagliflozin, dapagliflozin, and empagliflozin, were originally developed as glucose-lowering agents for type 2 diabetes. By blocking glucose reabsorption in the renal proximal tubules, these drugs induce glycosuria and improve various metabolic parameters [116]. Beyond their effects on glycemic control, SGLT2 inhibitors exhibit pleiotropic cardioprotective actions, such as enhancement of endothelial function and attenuation of oxidative stress, inflammation, and adverse myocardial remodeling [117,118].

Preclinical studies have demonstrated the cardioprotective potential of SGLT2 inhibitors in DIC [117]. Dapagliflozin, for example, reduces p65/NF-κB activation, preserves cardiac and renal tissue microstructure, and exerts systemic anti-inflammatory effects [117,119]. These drugs also mitigate oxidative stress and cardiomyocyte apoptosis, promote autophagy, and influence myocardial energy metabolism by enhancing ketone body utilization, thereby improving mitochondrial function [120,121,122]. Importantly, these cardioprotective effects do not interfere with the anti-cancer activity of anthracyclines [120].

SGLT2 inhibitors have demonstrated significant cardioprotective effects in clinical settings, with a class I indication for the treatment of HF, regardless of ejection fraction [123]. Building on preclinical evidence in DIC, emerging observational clinical studies suggest that SGLT2 inhibitors may also improve cardiovascular outcomes in cancer patients receiving anthracycline therapy [117,124,125]. Taken together, both preclinical and clinical data support the potential of SGLT2 inhibitors as cardioprotective agents, positioning this class of therapeutics as an important component of cardio-oncology strategies.

### 4.4. Pioglitazone

Pioglitazone is an oral antidiabetic agent from the thiazolidinedione (glitazone) class. It activates the peroxisome proliferator-activated receptor-gamma (PPAR-γ), a nuclear receptor that improves insulin’s effectiveness in regulating glucose metabolism [126]. Beyond its metabolic effects, pioglitazone has been linked to favorable cardiovascular effects, such as improved lipid profile, reduced vascular inflammation, and attenuation of atherosclerosis progression. However, its use is limited by adverse effects, particularly fluid retention and the increased risk of HF [127].

Thiazolidinediones have been shown to improve LV function in mouse models of HF following myocardial infarction (MI) [128] and to reduce the risk of major adverse cardiovascular events in patients with insulin resistance [129]. Moreover, one study demonstrated that pioglitazone exerted a protective effect against DOX-induced nephropathy in rats by mitigating profibrotic and inflammatory mechanisms [130]. Premedication with pioglitazone partially prevents LV dysfunction in both the acute and chronic phases in mice treated with DOX, suggesting that it may be a novel therapeutic strategy for cardioprotection in DIC. Nevertheless, the molecular basis of pioglitazone’s cardioprotective effects in DOX-induced LV dysfunction is still poorly understood [131].

### 4.5. Fibrates

Recent studies have highlighted the pleiotropic effects of fenofibrate, a PPARα activator, that provide direct myocardial protection in addition to its lipid-lowering effects [132]. Short-term treatment with fenofibrate was shown to improve vascular endothelial function in healthy, normolipidemic adults by reducing oxidative stress and enhancing eNOS activation [133]. Moreover, fenofibrate demonstrated beneficial effects in patients with systolic HF [134], while PPARα activation attenuated ET-1-induced cardiomyocyte hypertrophy [135].

Fenofibrate treatment provided cardioprotection in a mouse model of DIC. This was associated with increased circulating levels of endothelial progenitor cells, activation of cardiac NO signaling and angiogenesis pathways, and a reduced inflammatory response, all of which contributed to improved LV function. In addition, fenofibrate attenuated the DOX-induced increase in circulating BNP and pro-BNP [136].

A randomized controlled trial in breast cancer patients receiving DOX-based chemotherapy demonstrated that fenofibrate was safe, well-tolerated, and significantly reduced the incidence of DIC. Patients treated with fenofibrate had higher LV ejection fraction and reduced levels of N-terminal pro-BNP and myeloperoxidase compared to the control group [137].

### 4.6. Minocycline

Minocycline is a semisynthetic tetracycline antibiotic, widely used to treat a wide range of bacterial infections [138]. Many studies have shown that it can exert additional effects, including increased activity of antioxidant enzymes, suppression of proinflammatory cytokine production, inhibition of macrophage activation, and reduction in pro-apoptotic proteins [139].

Through these mechanisms, minocycline may mitigate DIC by lowering lipid peroxidation and inflammatory cytokine levels, while boosting the activity of antioxidant enzymes. Treatment with minocycline also alleviated the DOX-induced structural damage in cardiac tissue, including edema, cytoplasmic vacuolization, and inflammation. Despite robust evidence from in vitro and in vivo studies supporting its cardioprotective potential, clinical data remain limited [140].

### 4.7. Erythropoietin

Erythropoietin is classically known as a hematopoietic cytokine regulating erythropoiesis via its receptor, which is also expressed in non-hematopoietic tissues, including the heart [141]. Beyond its hematopoietic role, erythropoietin exerts cardiovascular protective effects against MI [142], ischemia-reperfusion injury [143], and DIC [144].

Collectively, preclinical studies in vitro and in vivo, as well as some clinical observations, support a cardioprotective role of erythropoietin against DOX-induced cardiac injury, mediated through the enhancement of mitochondrial biogenesis and function [145], modulation of extracellular matrix turnover and cardiac remodeling [146], and activation of the PI3K-Akt (Phosphatidylinositol 3-kinase (PI3K)/protein kinase B (Akt)) prosurvival signaling pathway [147].

### 4.8. Potential Repurposing for Target-Based Doxorubicin-Induced Cardiotoxicity Protection

Beyond classical repurposing, several new pharmacological strategies have been developed that directly target key mediators of DIC. Recent studies have focused on developing isoform-selective topoisomerase IIβ (TOP2β) inhibitors. These advances aim to further enhance cardioprotection [148]. A novel obex class of TOP2 inhibitors has been designed to target a distinct binding pocket in the ATPase domain of TOP2. Within this class, topobexin was engineered to interact with amino acid residues that differ between TOP2α and TOP2β. This confers selective inhibition of TOP2β with superior efficacy compared to dexrazoxane. In preclinical models, topobexin demonstrated potent protection against chronic anthracycline-induced cardiotoxicity [149]. Complementary to pharmacological strategies, siRNA-mediated knockdown of TOP2β has also been shown to mitigate DOX-induced DNA damage and apoptosis in cardiomyocytes [150].

Similarly, iron chelators (deferoxamine), ferroptosis inhibitors (ferrostatin-1, liproxstatin-1), and mitochondria-targeted antioxidants (MitoQ, MitoTEMPO) are being investigated as experimental approaches to reduce lipid peroxidation and mitochondrial injury in cardiomyocytes [32]. While these compounds are not yet suitable for repurposing, they exemplify future directions in mechanism-based cardioprotection.

In contrast, several clinically available drugs and natural compounds can already be considered as repurposing candidates. GPCR-targeting agents such as melatonin, ghrelin, and galanin analogs have been shown to reduce oxidative stress, apoptosis, and functional decline in preclinical models of DIC [151]. Melatonin protects against DOX-induced injury in mice and rats by reducing lipid peroxidation and exerting antioxidant effects, thereby inhibiting necrosis and apoptosis without diminishing its antitumor efficacy [152]. In rats and mice, cannabidiol protects against DOX-induced cardiac injury by reducing ROS/RNS accumulation, preserving mitochondrial function and biogenesis, enhancing cell survival, and suppressing myocardial inflammation [153,154]. Natural compounds such as curcumin [155], allicin [156], resveratrol [157], and (epi)catechin [158] exert protective effects by modulating GPCR signaling, including GPR97, GRK2, estrogen-related GPCRs, and adenosine receptors, thereby attenuating inflammation, oxidative stress, and maladaptive remodeling [151]. Moreover, stimulation of the apelinergic system (apelin/APJ receptor axis) has been shown to preserve cardiac function, improve vascular homeostasis, and reduce apoptosis in models of DOX-induced injury [159].

In addition, other drugs with established clinical use have been investigated for their potential to protect against DIC through specific molecular targets. For example, elamipretide stabilizes mitochondrial cardiolipin [160], bosentan and ambrisentan modulate endothelin receptors [161], doxycycline inhibits MMP-2, attenuates the LV cardiomyocyte hypertrophy and collagen type I expression [162], and experimental sodium–calcium exchanger inhibitors such as SEA0400 prevent calcium overload in cardiomyocytes [163]. Early preclinical studies suggest these agents may reduce oxidative stress, apoptosis, and maladaptive remodeling, supporting their potential for repurposing.

## 5. Paroxetine in Doxorubicin-Induced Cardiotoxicity Protection

Paroxetine is an antidepressant and the most potent selective inhibitor of serotonin reuptake (SSRI). SSRIs are the mainstay of treatment for psychological disorders that frequently occur in oncology patients, affecting both quality of life and life expectancy [164]. Paroxetine has also been shown to exert cardioprotective effects through GRK2 inhibition [165], while additionally demonstrating antiplatelet activity [166] and even antitumor properties [167].

The main disadvantage of paroxetine that needs to be addressed here is the risk to cardiovascular health. Conflicting results have been published on the ability of paroxetine and congeners to induce QT prolongation and life-threatening arrhythmias [168,169]. Based on recent literature, the risk of QT/QTc prolongation with the majority of newer non-SSRI antidepressants, including paroxetine, at therapeutic doses, is low [169]. In one real-life condition study, it was demonstrated that QT prolongation is not an SSRI class effect except for citalopram and escitalopram [168]. To further clarify the effect of paroxetine on cardiac electric activity, in an experimental study using the patch clamp technique on human voltage-gated sodium channels and by studying action potential on isolated rabbit LV cardiomyocytes, it was shown that paroxetine acts as an inhibitor of voltage-gated sodium channels, slows down conduction, and reduces cardiomyocyte excitability. It was concluded that this effect of paroxetine on sodium channels may be life-threatening and enhance the loss-of-function mutation in the *SCN5A* gene that underlies Brugada and long QT syndrome type 3 [170]. Also, caution is needed when paroxetine is combined with other drugs that can have synergistic effects on cardiac conduction and excitability or increase its bioavailability [171].

Another drawback of paroxetine is its effect on cardiac development and angiogenesis. In a study on isolated cardiomyocytes, paroxetine has been shown to disrupt mitochondrial function and reduce respiration and ATP production [172]. This embryotoxic effect was also diagnosed in humans. A meta-analysis confirmed a 28% increase in the risk of major cardiac malformations associated with paroxetine exposure during the first trimester of pregnancy [173]. Moreover, paroxetine was reported to be cardiotoxic to zebrafish larvae by triggering severe cardiac inflammation [174], additionally raising environmental safety concerns.

Nevertheless, under pathophysiological conditions, paroxetine has been reported to exert cardioprotection. This has been shown in experimental models of MI [175], DIC [15], primary hypertension [176], and pulmonary hypertension [177]. In cardiomyocytes, GRK2 is a pivotal signaling hub. It can integrate different transduction cascades, including key mediators of cardiovascular function such as catecholamines and angiotensin II. GRK2 also affects mitochondrial function by influencing processes like oxidative phosphorylation, ROS production, and apoptosis. GRK2 is up-regulated in cardiovascular diseases such as HF, cardiac hypertrophy, and hypertension [178].

In HF, maladaptive GRK2 up-regulation is triggered by excessive sympathetic activity. This causes downregulation of β-ARs, reducing the cardiac inotropic reserve. In an experimental MI model, pre-treatment of Wistar rats with paroxetine protected against post-MI cardiac remodeling. It reduced the expression of key genes mediating fibrosis, inflammation, and angiogenesis. Paroxetine’s effects were similar to those of βAR-antagonists and angiotensin II receptor type 1 (AT1) antagonists [175]. However, a single-center human retrospective study showed that short-term administration of paroxetine to hospitalized HF patients did not improve 28-day survival [179]. Still, a multicenter prospective randomized blinded study with long-term treatment would provide more reliable insight.

We investigated the potential of paroxetine to mitigate DIC in an established experimental model in rats. We found that DOX induces both phenotypes of HF, hypertrophic with preserved ejection fraction and dilated with reduced ejection fraction [15,49]. Paroxetine treatment significantly attenuated LV remodeling in both phenotypes, preserved myocardial structure, and improved cardiac function in rats treated with DOX. Importantly, it also improved overall survival compared with DOX-treated controls [15]. The mechanism by which paroxetine acts to reduce DOX toxicity seems to be complex and involves canonical and noncanonical effects of GRK2 inhibition, effects on inflammation, and oxidative stress [178].

Analysis of cardiac tissue by RT-qPCR and immunohistochemistry revealed downregulation and reduced synthesis of GRK2 and GRK3, alongside upregulation and enhanced synthesis of β1-AR and β2-AR. This increased β-AR expression may protect against maladaptive β-AR uncoupling and underlie the preservation of LV inotropic reserve observed with paroxetine treatment [15].

Downregulation of GRK2 by paroxetine in our experiments may have acted to reduce GRK2-mediated mitochondrial ROS formation and apoptosis [180,181], suggesting antioxidant properties [182] and contributing to the attenuation of DOX-induced cardiac injury. Another important beneficial effect of paroxetine is to block adrenaline-induced upregulation of fibrotic and hypertrophic genes such as connective tissue growth factor, collagen I, and beta-myosin heavy chain [176]. Downregulation of GRK2 could have also prevented nuclear factor of activated T-cells (NFAT) expression, a key cardiac hypertrophic factor [54].

These findings demonstrate that paroxetine can counteract both structural and functional cardiac damage induced by DOX, highlighting its potential as a repurposed agent for cardioprotection in oncology patients receiving high doses of DOX. The results provide a rationale for further preclinical and translational studies to elucidate the underlying molecular mechanisms and explore potential clinical applications.

## 6. Discussion and Future Directions

DIC remains a life-threatening complication of cancer therapy, with no effective preventive or therapeutic strategies currently available [17]. Drug repositioning is emerging as a promising strategy to mitigate DIC, taking advantage of existing safety data and potentially shortening the path and cost of translation into clinical practice [13]. According to our knowledge, this is the first comprehensive review addressing the principal repurposing candidates proposed for cardioprotection in the setting of DIC.

Antioxidant and anti-inflammatory actions are central to many promising candidates, including statins [98], SGLT2 inhibitors [124], and minocycline [140], which collectively reduce oxidative stress, modulate cytokine production, and preserve myocardial microstructure. PPAR activators, such as pioglitazone and fenofibrate, provide additional cardiovascular benefits through lipid-lowering effects and direct modulation of myocardial signaling pathways, improving endothelial function, reducing hypertrophy, and promoting angiogenesis [128,133]. Similarly, erythropoietin exerts cardioprotection by enhancing mitochondrial biogenesis, improving the balance of extracellular matrix turnover, and activating the PI3K-Akt survival pathway [141].

Paroxetine, in addition to anti-inflammatory and antioxidant properties, demonstrates cardioprotective effects in experimental models of DIC through modulation of GRK2/3 and β-AR signaling [15]. Downregulation of GRKs prevents maladaptive β-AR internalization and preserves LV inotropic reserve. This mechanism, alongside reported antiplatelet and antitumor properties of paroxetine [166,167], underscores its potential as a repurposed agent in cardio-oncology patients.

Despite these encouraging results, most evidence comes from animal studies or small clinical datasets. Optimal dosing regimens, timing of administration, and potential interactions with chemotherapy remain largely undefined. Moreover, phenotypic heterogeneity in DIC necessitates tailored strategies and the identification of predictive biomarkers to optimize patient selection for repurposed therapies.

Future perspectives should focus on combining drugs with different mechanisms, testing possible synergistic effects, and using pharmacogenomics to better tailor therapy. Advanced imaging, molecular profiling, and biomarker-guided strategies may enable early detection of subclinical DIC and facilitate timely intervention. Ultimately, well-designed randomized controlled trials are required to establish efficacy, safety, and clinical applicability in diverse cancer populations.

## 7. Conclusions

Drug repositioning is a promising strategy for cardioprotection in patients treated with DOX. Multiple agents with well-established safety profiles show potential to counteract oxidative stress, inflammation, maladaptive cardiac remodeling, and apoptosis, thereby preserving cardiac function. Among these, paroxetine stands out as a strong candidate, providing structural and functional protection in preclinical models through GRK2/3 modulation and preserved β-AR signaling. Based on current evidence, the preferential use of paroxetine instead of its SSRI congeners may represent a rational choice of antidepressant in oncology patients.

In the future, reducing DOX cardiotoxicity will likely require a multifaceted approach. Nanocarrier-based formulations are apparently the most promising way to increase the antitumor activity of DOX while reducing its cardiotoxicity; novel target-directed strategies remain promising but are largely preclinical. Drug repositioning, by contrast, offers a near-term opportunity, given the availability of agents with established safety. Well-designed, long-term clinical trials will be essential to balance oncologic efficacy with cardiovascular safety and enable more personalized treatment strategies.

## Figures and Tables

**Figure 1 ijms-26-10130-f001:**
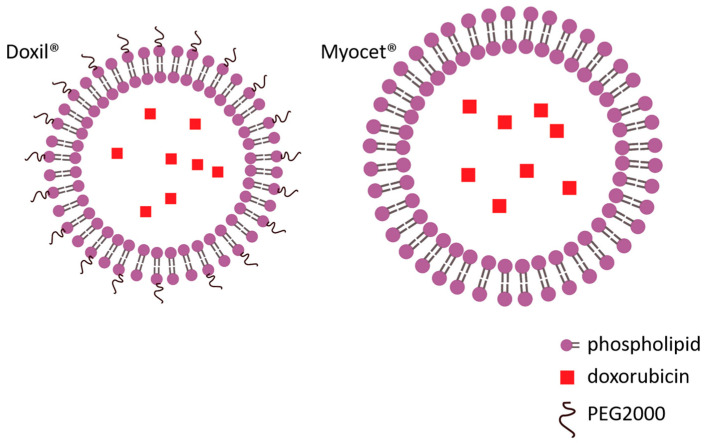
Liposomal formulations of doxorubicin: pegylated (Doxil^®^) and non-pegylated (Myocet^®^).

**Figure 2 ijms-26-10130-f002:**
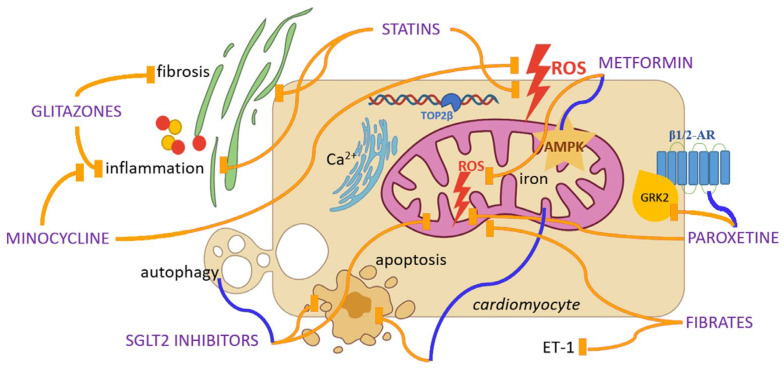
Repurposing candidates for cardioprotection in doxorubicin-induced cardiotoxicity and their potential mechanisms of action. Yellow lines indicate inhibition, while blue lines indicate activation. Abbreviations: ROS, Reactive Oxygen Species; TOP2β, Topoisomerase 2β; AMPK, Adenosine Monophosphate-Activated Protein Kinase; SGLT2, Sodium-Glucose Cotransporter-2; β-AR, Beta Adrenergic Receptor; GRK2, G protein-Coupled Receptor Kinase 2; ET-1, Endothelin-1.

**Table 1 ijms-26-10130-t001:** Repurposing candidates for cardioprotection in doxorubicin-induced cardiotoxicity.

Candidate Drug/Class	Primary Mechanism(s) of Action in DIC	Preclinical Evidence	Clinical Evidence	References
Statins	↓ ROS (NOX-1 inhibition, ↑ antioxidant enzymes), ↓ NF-κB inflammation, ↓ fibrosis, AMPK activation, ↑ eNOS	Strong, multiple animal and cellular models	Retrospective studies (↓ HF incidence in breast cancer patients); ESC guidelines recommend in high-risk patients	[98,99,100,101,102,103,104,105,106,107,108,109]
Metformin	AMPK activation, ↓ oxidative stress, ↑ mitochondrial function, normalization of autophagy	Robust in vitro and in vivo	Limited; one terminated phase II trial (NCT02472353)	[110,111,112,113,114,115]
SGLT2 inhibitors (dapagliflozin, empagliflozin, canagliflozin)	↓ inflammation (↓ NF-κB), ↓ oxidative stress, ↑ autophagy, improved energy metabolism (↑ ketone use), preserved microstructure	Strong preclinical data	Large HF trials (non-DIC); early observational data in cancer patients	[116,117,118,119,120,121,122,123,124,125]
Pioglitazone	PPAR-γ activation, ↓ inflammation, ↓ fibrosis, improved LV function	Positive in mouse models of DIC and MI	Limited; CVD trials outside oncology	[126,127,128,129,130,131]
Fibrates (fenofibrate)	PPAR-α activation, ↓ oxidative stress, ↑ eNOS, angiogenesis, ↓ BNP/pro-BNP, ↑ endothelial progenitors	Strong in mouse DIC models	RCT in breast cancer: safe, ↓ DIC incidence, improved EF	[132,133,134,135,136,137]
Minocycline	↑ antioxidant enzymes, ↓ lipid peroxidation, ↓ cytokines, ↓ apoptosis, protection against structural damage	Robust in vitro and in vivo	Very limited	[138,139,140]
Erythropoietin	↑ mitochondrial biogenesis, PI3K-Akt activation, ↓ remodeling/fibrosis	Consistent preclinical	Limited; some supportive observations	[141,142,143,144,145,146,147]

ROS, Reactive Oxygen Species; NOX, Nicotinamide Adenine Dinucleotide Phosphate Oxidase; NF-kB, Nuclear Factor Kappa B; AMPK, Adenosine Monophosphate-Activated Protein Kinase; eNOS, Endothelial Nitric Oxide Synthase; PPAR, Peroxisome Proliferator-Activated Receptor; LV, Left Ventricular; BNP, B-type Natriuretic Peptide; PI3K-Akt, Phosphatidylinositol 3-kinase (PI3K)/protein kinase B (Akt); HF, Heart Failure; ESC, European Society of Cardiology; CVD, Cardiovascular Disease; RCT, Randomized Clinical Trial; DIC, Doxorubicin-induced Cardiotoxicity; EF, Ejection Fraction.

## Data Availability

No new data were created or analyzed in this study. Data sharing is not applicable to this article.

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
