# Peer review of "Drug Repositioning in Doxorubicin-Induced Cardiotoxicity Protection"

_ijms, 2025, doi:10.3390/ijms262010130_

Round 1
Reviewer 1 Report
Comments and Suggestions for Authors
The authors present a review on drug repurposing for doxorubicin-induced cardiotoxicity (DIC), with a particular focus on the antidepressant paroxetine. The article presents a good review in terms of the quality, organization, and impact of information presented.
This work merits publication after minor corrections.
- Although the manuscript is well-written and well-organized, I would suggest improving the impact and soundness by adding the following:
Adding a part at the end (Potential repurposing for target-based DIC protection) that describes available drugs and those in clinical trials which modulate the protective targets that the authors have properly addressed in the review including (whenever possible); TOP2β, mitochondrial cardiolipin, NOX2, NOS, Iron chelators, Na⁺/Ca²⁺ exchanger, adenosine monophosphate-activated protein kinase (AMPK), endothelin A and B, MMP-2 and MMP-9, G protein-coupled receptor kinases (GRKs).

Author Response
The authors present a review on drug repurposing for doxorubicin-induced cardiotoxicity (DIC), with a particular focus on the antidepressant paroxetine. The article presents a good review in terms of the quality, organization, and impact of information presented.
This work merits publication after minor corrections.
- Although the manuscript is well-written and well-organized, I would suggest improving the impact and soundness by adding the following:
Adding a part at the end (Potential repurposing for target-based DIC protection) that describes available drugs and those in clinical trials which modulate the protective targets that the authors have properly addressed in the review including (whenever possible); TOP2β, mitochondrial cardiolipin, NOX2, NOS, Iron chelators, Na⁺/Ca²⁺ exchanger, adenosine monophosphate-activated protein kinase (AMPK), endothelin A and B, MMP-2 and MMP-9, G protein-coupled receptor kinases (GRKs).
Thank you very much for the suggestion. We have added a new section Potential Repurposing for Target-based Doxorubicin-Induced Cardiotoxicity Protection, summarizing available drugs. See page 17 (lines 1-44) and page 18 (1-13).

Reviewer 2 Report
Comments and Suggestions for Authors
The work of Marija Kosić, entitled Drug Repositioning in Doxorubicin-Induced Cardiotoxicity Protection, is an interesting review on drug repositioning and potential new therapies in doxorubicin-induced cardiotoxicity (DIC). However, the main text—particularly the section on physiopathology—requires restructuring, as several mini-paragraphs appear in the main text. In spite, it would be valuable to include in the figure the mechanisms associated with cardiotoxicity, integrating other agents described in the article, such as minocycline and fibrates. Furthermore, the review would benefit from a more detailed discussion of the role of epigenetic changes induced by doxorubicin, as well as an expanded section addressing fibrosis and inflammation as key contributors to DIC.
It is also recommended to reconsider the idea that cardiotoxicity may be secondary to effects on cell types beyond cardiomyocytes, such as fibroblasts (Patricelli, C., Lehmann, P., Oxford, J.T. et al. Doxorubicin-induced modulation of TGF-β signaling cascade in mouse fibroblasts: insights into cardiotoxicity mechanisms. Sci Rep 13, 18944, 2023. https://doi.org/10.1038/s41598-023-46216-7; Levick SP, Soto-Pantoja DR, Bi J, Hundley WG, Widiapradja A, Manteufel EJ, Bradshaw TW, Meléndez GC. Doxorubicin-Induced Myocardial Fibrosis Involves the Neurokinin-1 Receptor and Direct Effects on Cardiac Fibroblasts. Heart Lung Circ. 2019;28(10):1598–1605. doi: 10.1016/j.hlc.2018.08.003).
Although the authors attempt to provide a broad analysis of potential strategies to reduce cardiotoxicity, including titration and pharmacogenomics, this approach somewhat distracts from the initial objectives of the review. The section on paroxetine, in particular, should be restructured to maintain neutrality and the expected third-person academic style. In addition, the authors should further highlight their own previous contributions to the field.
Regarding paroxetine, it is important to note that treatment in patients with heart failure did not significantly reduce 28-day all-cause mortality (Xu H, Meng L, Long H, Shi Y, Liu Y, Wang L, Liu D. Paroxetine and Mortality in Heart Failure: A Retrospective Cohort Study. Front Cardiovasc Med. 2022. doi: 10.3389/fcvm.2021.794584). The discussion should also address the potential for cardiotoxicity, as observed in other species. For example, paroxetine has been shown to induce cardiotoxic effects in zebrafish larvae, where inflammation was proposed as a potential mechanism of cardiac dysfunction (Zhu Y, Song F, Gu J, Wu L, Wu W, Ji G. Paroxetine induced larva zebrafish cardiotoxicity through inflammation response. Ecotoxicol Environ Saf. 2023 Jul 15;260:115096. doi: 10.1016/j.ecoenv.2023). Additionally, paroxetine has been reported to slow conduction and reduce excitability in cardiac cells (Plijter, I.S.; Verkerk, A.O.; Wilders, R. The Antidepressant Paroxetine Reduces the Cardiac Sodium Current. Int. J. Mol. Sci. 2023, 24, 1904. https://doi.org/10.3390/ijms24031904). These are also relevant evidence that should be discussed in the section.
Moreover, the discussion should be expanded to cover evidence suggesting that paroxetine may modulate fibrotic, inflammatory, and angiogenesis-related pathways in post-myocardial infarction models (Alonazi AS, Almodawah S, Aldigi R, Bin Dayel A, Alamin M, Almotairi AR, El-Tohamy MF, Alharbi H, Ali R, Alshammari TK, Alrasheed NM. Potential cardioprotective effect of paroxetine against ventricular remodeling in an animal model of myocardial infarction: a comparative study. BMC Pharmacol Toxicol. 2024;25(1):99. doi: 10.1186/s40360-024-00824-9).
Finally, the inclusion of a summary table would be helpful to allow readers to easily identify the main beneficial outcomes of the pharmacological treatments discussed.
Author Response
The work of Marija Kosić, entitled Drug Repositioning in Doxorubicin-Induced Cardiotoxicity Protection, is an interesting review on drug repositioning and potential new therapies in doxorubicin-induced cardiotoxicity (DIC). However, the main text—particularly the section on physiopathology—requires restructuring, as several mini-paragraphs appear in the main text. In spite, it would be valuable to include in the figure the mechanisms associated with cardiotoxicity, integrating other agents described in the article, such as minocycline and fibrates. Furthermore, the review would benefit from a more detailed discussion of the role of epigenetic changes induced by doxorubicin, as well as an expanded section addressing fibrosis and inflammation as key contributors to DIC.
Thank you for your suggestion. The text has been restructured as you suggested. Minocyclin and fibrates have been included in Figure 2. The role of epigenetic changes induced by doxorubicine has been added on page 6 (lines 27-43) and page 7 (lines 1-7) as well as an expanded section addressing fibrosis and inflammation as contributors to DIC on page 5 (lines 25-44) and page 6 (lines 1-4).
It is also recommended to reconsider the idea that cardiotoxicity may be secondary to effects on cell types beyond cardiomyocytes, such as fibroblasts (Patricelli, C., Lehmann, P., Oxford, J.T. et al. Doxorubicin-induced modulation of TGF-β signaling cascade in mouse fibroblasts: insights into cardiotoxicity mechanisms. Sci Rep 13, 18944, 2023. https://doi.org/10.1038/s41598-023-46216-7; Levick SP, Soto-Pantoja DR, Bi J, Hundley WG, Widiapradja A, Manteufel EJ, Bradshaw TW, Meléndez GC. Doxorubicin-Induced Myocardial Fibrosis Involves the Neurokinin-1 Receptor and Direct Effects on Cardiac Fibroblasts. Heart Lung Circ. 2019;28(10):1598–1605. doi: 10.1016/j.hlc.2018.08.003).
The section about the role of fibroblasts and other cellular targets in DIC has been added on page 7 (lines 9-34). We include proposed references on page 5 (line 30) and page 6 (line 4).
Although the authors attempt to provide a broad analysis of potential strategies to reduce cardiotoxicity, including titration and pharmacogenomics, this approach somewhat distracts from the initial objectives of the review.
We agree; however, reviewers 1 and 3 requested expanded sections on pharmacokinetics (page 8 lines 22-33) and epigenetics (page 6 lines 27-43 and page 7 lines 1-7). We hope we found a reasonable measure to meet all suggestions.
The section on paroxetine, in particular, should be restructured to maintain neutrality and the expected third-person academic style. In addition, the authors should further highlight their own previous contributions to the field.
Regarding paroxetine, it is important to note that treatment in patients with heart failure did not significantly reduce 28-day all-cause mortality (Xu H, Meng L, Long H, Shi Y, Liu Y, Wang L, Liu D. Paroxetine and Mortality in Heart Failure: A Retrospective Cohort Study. Front Cardiovasc Med. 2022. doi: 10.3389/fcvm.2021.794584). The discussion should also address the potential for cardiotoxicity, as observed in other species. For example, paroxetine has been shown to induce cardiotoxic effects in zebrafish larvae, where inflammation was proposed as a potential mechanism of cardiac dysfunction (Zhu Y, Song F, Gu J, Wu L, Wu W, Ji G. Paroxetine induced larva zebrafish cardiotoxicity through inflammation response. Ecotoxicol Environ Saf. 2023 Jul 15;260:115096. doi: 10.1016/j.ecoenv.2023). Additionally, paroxetine has been reported to slow conduction and reduce excitability in cardiac cells (Plijter, I.S.; Verkerk, A.O.; Wilders, R. The Antidepressant Paroxetine Reduces the Cardiac Sodium Current. Int. J. Mol. Sci. 2023, 24, 1904. https://doi.org/10.3390/ijms24031904). These are also relevant evidence that should be discussed in the section.
Thank you for this observation. The section on paroxetine is restructured to maintain neutrality and a third-person academic style. We added a paragraph on cardiovascular risks, embryotoxicity, and environmental concerns, and we included all the suggested publications. See page 20 (lines 10-38) and page 21 (lines 12-16).
Moreover, the discussion should be expanded to cover evidence suggesting that paroxetine may modulate fibrotic, inflammatory, and angiogenesis-related pathways in post-myocardial infarction models (Alonazi AS, Almodawah S, Aldigi R, Bin Dayel A, Alamin M, Almotairi AR, El-Tohamy MF, Alharbi H, Ali R, Alshammari TK, Alrasheed NM. Potential cardioprotective effect of paroxetine against ventricular remodeling in an animal model of myocardial infarction: a comparative study. BMC Pharmacol Toxicol. 2024;25(1):99. doi: 10.1186/s40360-024-00824-9).
Effects of paroxetine on inflammation, fibrosis, and angiogenesis, including the suggested reference, are added on page 20 (lines 39-44) and page 21 (lines 1-12).
Finally, the inclusion of a summary table would be helpful to allow readers to easily identify the main beneficial outcomes of the pharmacological treatments discussed.
Thank you for suggesting. A summary table on repurposed drugs has been added (Page 19).

Reviewer 3 Report
Comments and Suggestions for Authors
Although doxorubicin is one of the most long-used drugs in tumor chemotherapy, its versatile antitumor activity has traditionally attracted attention to its use. At the same time, the use of doxorubicin in tumor therapy is associated with risks caused by side cardiotoxic effects. Therefore, when treating tumors with doxorubicin, it is important to maintain a balance between the dose and duration of drug administration and the achieved therapeutic effect. This complex problem is addressed in this review, which can be rated quite highly. However, some corrections must be made before its publication:
1. It would be good to indicate the time period for which the literature data are analyzed in the introduction, as well as to note the main literature reviews conducted earlier in this area. Also, show how this review compares favorably with previous ones.
2. The review is practically not illustrated. Add figures that could attract the attention of a potential reader. 3. Pay more attention to pharmacokinetics when describing the therapeutic possibilities and limitations of tumor chemotherapy using doxorubicin.
4. The review does not note the latest advances in reducing doxorubicin cardiotoxicity using nanosized particles. This problem should be considered in detail, since it is covered in a number of sources. For example, in the works of A.M. Nechaeva (Scopus ID: 57226606676) and many other articles. Apparently, the use of nanosized delivery systems is the most promising way to increase the antitumor activity of doxorubicin while reducing its cardiotoxicity. Therefore, it would be good to consider these important points in the review.
5. The review is replete with references to literature older than 10 years. Please add more new literature references for the last 5 years. In addition, 126 references are too few for this area, where a large number of works have been carried out.
6. The conclusion does not provide the authors' opinion on the prospects for reducing the side cardiotoxicity of doxorubicin. The "Conclusion" section could do with some expansion.
Author Response
Although doxorubicin is one of the most long-used drugs in tumor chemotherapy, its versatile antitumor activity has traditionally attracted attention to its use. At the same time, the use of doxorubicin in tumor therapy is associated with risks caused by side cardiotoxic effects. Therefore, when treating tumors with doxorubicin, it is important to maintain a balance between the dose and duration of drug administration and the achieved therapeutic effect. This complex problem is addressed in this review, which can be rated quite highly. However, some corrections must be made before its publication:
- It would be good to indicate the time period for which the literature data are analyzed in the introduction, as well as to note the main literature reviews conducted earlier in this area. Also, show how this review compares favorably with previous ones.
Thank you for noticing. We have now added a sentence addressing your remark. The literature search on doxorubicin-induced cardiotoxicity in PubMed revealed 297 review articles published over the past five years (https://pubmed.ncbi.nlm.nih.gov/?term=doxorubicin+cardiotoxicity). It was impossible to cite all of them, but, if you agree, we provided a link to the full list. We also cited the first original contributions to this field, which date back more than five years. To the best of our knowledge, this is the first comprehensive review that integrates the mechanisms of DIC with the potential role of drug repurposing, with a particular focus on paroxetine (see page 2, lines 35-40).
The review is practically not illustrated. Add figures that could attract the attention of a potential reader.
The existing figure integrates main toxicological mechanisms and repurposed drugs, in order to avoid too busy figures and redundancy and keeping in mind the scope of this review. However, since doxorubicin incapsulated in liposomal nanoparticles has most effectively reduced cardiotoxicity, we have now added an illustration (Figure 1) on them. We have also added a table summarizing repurposed drugs in DIC (see page 19).
- Pay more attention to pharmacokinetics when describing the therapeutic possibilities and limitations of tumor chemotherapy using doxorubicin.
We have expanded the section on pharmacokinetics See page 8 lines 22-33.
- The review does not note the latest advances in reducing doxorubicin cardiotoxicity using nanosized particles. This problem should be considered in detail, since it is covered in a number of sources. For example, in the works of A.M. Nechaeva (Scopus ID: 57226606676) and many other articles. Apparently, the use of nanosized delivery systems is the most promising way to increase the antitumor activity of doxorubicin while reducing its cardiotoxicity. Therefore, it would be good to consider these important points in the review.
The section on doxorubicin nanoparticles and a figure has been added, as well as the suggested reference. See page 9 lines 1-22.
- The review is replete with references to literature older than 10 years. Please add more new literature references for the last 5 years. In addition, 126 references are too few for this area, where a large number of works have been carried out.
Thank you for noticing, we have added newer references now. In total, the number of references is 183. We also indicate a link to the latest review articles, which, due to a number cannot all be cited (page 2 line 40). We also cited unavoidable first original contributions to the field, which are far back than 5 years.
- The conclusion does not provide the authors' opinion on the prospects for reducing the side cardiotoxicity of doxorubicin. The "Conclusion" section could do with some expansion.
The conclusion has been expanded by a few sentences. See page 23 lines 16-24.

Round 2
Reviewer 2 Report
Comments and Suggestions for Authors
The authors have satisfactorily addressed the raised comments and questions, significantly enhancing the understanding of the manuscript's initial limitations and inquiries.
Author Response
Comment: The authors have satisfactorily addressed the raised comments and questions, significantly enhancing the understanding of the manuscript's initial limitations and inquiries.
Reply: thank you very much for your suggestions that have significantly improved our manuscript
Reviewer 3 Report
Comments and Suggestions for Authors
The authors have adequately revised the manuscript and I recommend its acceptance in its current form.
Author Response
comment: The authors have adequately revised the manuscript and I recommend its acceptance in its current form.
response: thank you for approval